# Differentiating the learning styles of college students in different disciplines in a college English blended learning setting

**Jie Hu** [1,2,3]*, **Yi Peng** [1], **Xueliang Chen** [1], **Hangyan Yu** [1]

**1** Department of Linguistics, School of International Studies, Zhejiang University, Hangzhou City, Zhejiang Province, China, **2** Center for College Foreign Language Teaching, Zhejiang University, Hangzhou City, Zhejiang Province, China, **3** Institute of Asian Civilizations, Zhejiang University, Hangzhou City, Zhejiang Province, China

* huj@zju.edu.cn

**Data Availability Statement:** All relevant data are within the paper and its Supporting Information files.

## Abstract

Learning styles are critical to educational psychology, especially when investigating various contextual factors that interact with individual learning styles. Drawing upon Biglan's taxonomy of academic tribes, this study systematically analyzed the learning styles of 790 sophomores in a blended learning course with 46 specializations using a novel machine learning algorithm called the support vector machine (SVM). Moreover, an SVM-based recursive feature elimination (SVM-RFE) technique was integrated to identify the differential features among distinct disciplines. The findings of this study shed light on the optimal feature sets that collectively determined students' discipline-specific learning styles in a college blended learning setting.

## Introduction

### Research background

Learning style, as an integral and vital part of a student's learning process, has been constantly discussed in the field of education and pedagogy. Originally developed from the field of psychology, psychological classification, and cognitive research several decades ago [1], the term "learning style" is generally defined as the learner's innate and individualized preference for ways of participation in learning practice [2]. Theoretically, learning style provides a window into students' learning processes [3, 4], predicts students' learning outcomes [5, 6], and plays a critical role in designing individualized instruction [7]. Knowing a student's learning style and personalizing instruction to students' learning style could enhance their satisfaction [8], improve their academic performance [9], and even reduce the time necessary to learn [10].

Researchers in recent years have explored students' learning styles from various perspectives [11–13]. However, knowledge of the learning styles of students from different disciplines in blended learning environments is limited. In an effort to address this gap, this study aims to achieve two major objectives. First, it investigates how disciplinary background impacts students' learning styles in a blended learning environment based on data collected in a

**Funding:** This research was supported by the Philosophical and Social Sciences Planning Project of Zhejiang Province in 2020 [grant number 20NDJC01Z] with the recipient Jie Hu, Second Batch of 2019 Industry-University Collaborative Education Project of Chinese Ministry of Education [grant number 201902016038] with the recipient Jie Hu, SUPERB College English Action Plan with the recipient Jie Hu, and the Fundamental Research Funds for the Central Universities of Zhejiang University with the recipient Jie Hu.

**Competing interests:** The authors have declared that no competing interests exist.

compulsory college English course. Students across 46 disciplines were enrolled in this course, providing numerous disciplinary factor resources for investigating learning styles. Second, it introduces a novel machine learning method named the SVM to the field of education to identify an optimal set of factors that can simultaneously differentiate students of different academic disciplines. Based on data for students from 46 disciplines, this research delves into the effects of a massive quantity of variables related to students' learning styles with the help of a powerful machine learning algorithm. Considering the convergence of a wide range of academic disciplines and the detection of latent interactions between a large number of variables, this study aims to provide a clear picture of the relationship between disciplinary factors and students' learning styles in a blended learning setting.

## Literature review

**Theories of learning styles.** Learning style is broadly defined as the inherent preferences of individuals as to how they engage in the learning process [2], and the "cognitive, affective and physiological traits" of students have received special attention [14]. To date, there has been a proliferation of learning style definitions proposed to explain people's learning preferences, each focusing on different aspects. Efforts to dissect learning style have been contested, with some highlighting the dynamic process of the learner's interaction with the learning environment [14] and others underlining the individualized ways of information processing [15]. One vivid explication involved the metaphor of an onion, pointing out the multilayer nature of learning styles. It was proposed that the outermost layer of the learning style could change in accordance with the external environment, while the inner layer is relatively stable [16, 17]. In addition, a strong concern in this field during the last three decades has led to a proliferation of models that are germane to learning styles, including the Kolb model [18], the Myers-Briggs Type Indicator model [19] and the Felder-Silverman learning style model (FSLSM) [20]. These learning style models have provided useful analytical lenses for analyzing students' learning styles. The Kolb model focuses on learners' thinking processes and identifies four types of learning, namely, diverging, assimilating, converging, and accommodating [18]. The Myers-Briggs Type Indicator model classifies learners into extraversion and introversion types, with the former preferring to learn from interpersonal communication and the latter inclining to benefit from personal experience [19]. As the most popular available model, the FSLSM identifies eight categories of learners according to the four dimensions of perception, input, processing and understanding [20]. In contrast to other learning style models that divided students into only a few groups, the FSLSM describes students' learning styles in a more detailed manner. The four paired dimensions delicately distinguish students' engagement in the learning process, providing a solid basis for a steady and reliable learning style analysis [21]. In addition, it has been argued that the FSLSM is the most appropriate model for a technology-enhanced learning environment because it involves important theories of cognitive learning behaviors [22, 23]. Therefore, a large number of scholars have based their investigations of students' learning styles in the e-learning/computer-aided learning environment on FSLSM [24–28].

**Learning styles and FSLSM.** Different students receive, process, and respond to information with different learning styles. A theoretical model of learning style can be used to categorize people according to their idiosyncratic learning styles. In this study, the FSLSM was adopted as a theoretical framework to address the collective impacts of differences in students' learning styles across different disciplines (see Fig 1).

**The FSLSM includes learning styles scattered among four dimensions.** Visual learners process information best when it is presented as graphs, pictures, etc., while verbal learners

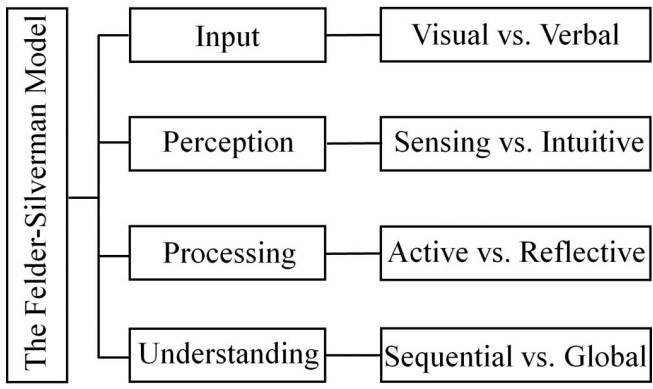

**Fig 1. The adapted Felder-Silverman learning style model.** This model specifies the four dimensions of the construct of learning style: visual/verbal, sensing/intuitive, active/reflective, and sequential/global. These four dimensions correspond to four psychological processes: input, perception, processing, and understanding.

prefer spoken cues and remember best what they hear. Sensory learners like working with facts, data, and experimentation, while intuitive learners prefer abstract principles and theories. Active learners like to try things and learn through experimentation, while reflective learners prefer to think things through before taking action. Sequential learners absorb knowledge in a linear fashion and make progress step by step, while global learners tend to grasp the big picture before filling in all the details.

**Learning styles and academic disciplines.** Learning styles vary depending on a series of factors, including but not limited to age [29], gender [30], personality [2, 31], learning environment [32] and learning experience [33]. In the higher education context, the academic discipline seems to be an important variable that influences students' distinctive learning styles, which echoes a multitude of investigations [29, 34–41]. One notable study explored the learning styles of students from 4 clusters of disciplines in an academic English language course and proposed that the academic discipline is a significant predictor of students' learning styles, with students from the soft-pure, soft-applied, hard-pure and hard-applied disciplines each favoring different learning modes [42]. In particular, researchers used the Inventory of Learning Styles (ILS) questionnaire and found prominent disparities in learning styles between students from four different disciplinary backgrounds in the special educational field of vocational training [43]. These studies have found significant differences between the learning styles of students from different academic disciplines, thus supporting the concept that learning style could be domain dependent.

**Learning styles in an online/blended learning environment.** Individuals' learning styles reflect their adaptive orientation to learning and are not fixed personality traits. Consequently, learning styles can vary among diverse contexts, and related research in different contexts is vital to understanding learning styles in greater depth. Web-based technologies eliminate barriers of space and time and have become integrated in individuals' daily lives and learning habits. Online and blended learning have begun to pervade virtually every aspect of the education landscape [40], and this warrants close attention. In addition to a series of studies that reflected upon the application of information and communication technology in the learning process [44, 45], recent studies have found a mixed picture of whether students in a web-based/blended learning environment have a typical preference for learning.

Online learning makes it possible for students to set their goals and develop an individualized study plan, equipping them with more learning autonomy [46]. Generally, students with a more independent learning style, greater self-regulating behavior and stronger self-efficacy are

found to be more successful in an online environment [47]. For now, researchers have made substantial contributions to the identification and prediction of learning styles in an online learning environment [27, 48–51]. For instance, an inspiring study focused on the manifestation of college students' learning styles in a purely computer-based learning environment to evaluate the different learning styles of web-learners in the online courses, indicating that students' learning styles were significantly related to online participation [49]. Students' learning styles in interactive E-learning have also been meticulously investigated, from which online tutorials have been found to be contributive to students' academic performance regardless of their learning styles [51].

As a flexible learning method, blended courses have combined the advantages of both online learning and traditional teaching methods [52]. Researchers have investigated students' learning styles within this context and have identified a series of prominent factors, including perceived satisfaction and technology acceptance [53], the dynamics of the online/face-to-face environment [54], and curriculum design [55]. Based on the Visual, Aural, Reading or Write and Kinesthetic model, a comprehensive study scrutinized the learning styles of K12 students in a blended learning environment, elucidating the effect of the relationship between personality, learning style and satisfaction on educational outcomes [56]. A recent study underscored the negative effects of kinesthetic learning style, whereas the positive effects of visual or auditory learning styles on students' academic performance, were also marked in the context of blended learning [57].

Considering that academic disciplines and learning environment are generally regarded as essential predictors of students' learning styles, some studies have also concentrated on the effects of academic discipline in a blended learning environment. Focusing on college students' learning styles in a computer-based learning environment, an inspiring study evaluated the different learning styles of web learners, namely, visual, sensing, global and sequential learners, in online courses. According to the analysis, compared with students from other colleges, liberal arts students, are more susceptible to the uneasiness that may result from remote teaching because of their learning styles [11]. A similar effort was made with the help of the CMS tool usage logs and course evaluations to explore the learning styles of disciplinary quadrants in the online learning environment. The results indicated that there were noticeable differences in tool preferences between students from different domains [12]. In comparison, within the context of blended learning, a comprehensive study employed chi-square statistics on the basis of the Community of Inquiry (CoI) presences framework, arguing that soft-applied discipline learners in the blended learning environment prefer the kinesthetic learning style, while no correlations between the learning style of soft-pure and hard-pure discipline students and the CoI presences were identified. However, it is noted that students' blended learning experience depends heavily on academic discipline, especially for students in hard-pure disciplines [13].

## Research gaps and research questions

Overall, the research seems to be gaining traction, and new perspectives are continually introduced. The recent literature on learning styles mostly focuses on the exploration of the disciplinary effects on the variation in learning styles, and some of these studies were conducted within the blended environment. However, most of the studies focused only on several discrete disciplines or included only a small group of student samples [34–41]. Data in these studies were gathered through specialized courses such as academic English language [42] rather than the compulsory courses available to students from all disciplines. Even though certain investigations indeed boasted a large number of samples [49], the role of teaching was emphasized rather than students' learning style. In addition, what is often overlooked is that a large

number of variables related to learning styles could distinguish students from different academic disciplines in a blended learning environment, whereas a more comprehensive analysis that takes into consideration the effects of a great quantity of variables related to learning styles has remained absent. Therefore, one goal of the present study is to fill this gap and shed light on this topic.

Another issue addressed in this study is the selection of an optimal measurement that can effectively identify and differentiate individual learning styles [58]. The effective identification and differentiation of individual learning styles can not only help students develop greater awareness of their learning but also provide teachers with the necessary input to design tailor-made instructions in pedagogical practice. Currently, there are two general approaches to identify learning styles: a literature-based approach and a data-driven approach. The literature-based approach tends to borrow established rules from the existing literature, while the data-driven approach tends to construct statistical models using algorithms from fields such as machine learning, artificial intelligence, and data mining [59]. Research related to learning styles has been performed using predominantly traditional instruments, such as descriptive statistics, Spearman's rank correlation, coefficient R [39], multivariate analysis of variance [56] and analysis of variance (ANOVA) [38, 43, 49, 57]. Admittedly, these instruments have been applied and validated in numerous studies, in different disciplines, and across multiple time-scales. Nevertheless, some of the studies using these statistical tools did not identify significant results [36, 53, 54] or reached only loose conclusions [60]; this might be because of the inability of these methods to probe into the synergistic effects of variables. However, the limited functions of comparison, correlation, prediction, etc. are being complemented by a new generation of technological innovations that promise more varied approaches to addressing social and scientific issues. Machine learning is one such approach that has received much attention both in academia and beyond. As a subset of artificial intelligence, machine learning deals with algorithms and statistical models on computer systems, performing tasks based on patterns and inference instead of explicit instruction. As such, it can deal with high volumes of data at the same time, perform tasks automatically and independently, and continuously improve its performance based on past experience [54]. Similar machine learning approaches have been proposed and tested by different scholars to identify students' learning styles, with varying results regarding the classification of learning styles. For instance, a study that examined the precision levels of four computational intelligence approaches, i.e., artificial neural network, genetic algorithm, ant colony system and particle swarm optimization, found that the average precision of learning style differentiation ranged between 66% and 77% [61]. Another study that classified learning styles through SVM reported accuracy levels ranging from 53% to 84% [62]. A comparison of the prediction performance of SVM and artificial neural networks found that SVM has higher prediction accuracy than the latter [63]. This was further supported by another study, which yielded a similar result between SVM and the particle swarm optimization algorithm [64]. Moreover, when complemented by a genetic algorithm [65] and ant colony system [66], SVM has also shown improved results. These findings across different fields point to the reliability of SVM as an effective statistical tool for identification and differentiation analysis.

Therefore, a comprehensive investigation across the four general disciplines in Biglan's taxonomy using a strong machine learning approach is needed. Given the existence of the research gaps discussed above, this exploratory study seeks to address the following questions:

1. Can students' learning styles be applied to differentiate various academic disciplines in the blended learning setting? If so, what are the differentiability levels among different academic disciplines based on students' learning styles?

2. What are the key features that can be selected to determine the collective impact on differentiation by a machine learning algorithm?

3. What are the collective impacts of optimal feature sets?

## Materials and methods

This study adopted a quantitative approach for the analysis. First, a modified and translated version of the original ILS questionnaire was administered to collect scores for students' learning styles. Then, two alternate data analyses were performed separately. One analysis involved a traditional ANOVA, which tested the main effect of discipline on students' learning styles in each ILS dimension. The other analysis involved the support vector machine (SVM) technique to test its performance in classifying students' learning styles in the blended learning course among 46 specializations. Then, SVM-based recursive feature elimination (SVM-RFE) was employed to specify the impact of students' disciplinary backgrounds on their learning styles in blended learning. By referencing the 44 questions (operationalized as features in this study) in the ILS questionnaire, SVM-RFE could rank these features based on their relative importance in differentiating different disciplines and identify the key features that collectively differentiate the students' learning style. These steps are intended to not only identify students' learning style differences but also explain such differences in relation to their academic disciplinary backgrounds.

### Participants

The participants included 790 sophomores taking the blended English language course from 46 majors at Z University. Sophomore students were selected for this study for two reasons. First, sophomores are one of the only two groups of students (the other group being college freshmen) who take a compulsory English language course, namely, the College English language course. Second, of these two groups of students, sophomores have received academic discipline-related education, while their freshmen counterparts have not had disciplinary training during the first year of college. In the College English language course, online activities, representing 55% of the whole course, include e-course teaching designed by qualified course teachers or professors, courseware usage for online tutorials, forum discussion and essay writing, and two online quizzes. Offline activities, which represent 45% of the whole course, include role-playing, ice-breaker activities, group presentations, an oral examination, and a final examination. Therefore, the effects of the academic discipline on sophomores' learning styles might be sufficiently salient to warrant a comparison in a blended learning setting [67]. Among the participants, 420 were male, and 370 were female. Most participants were aged 18 to 19 years and had taken English language courses for at least 6 years. Based on Biglan's typology of disciplinary fields, the students' specializations were classified into the four broad disciplines of hard-applied (HA, 289/37.00%), hard-pure (HP, 150/19.00%), soft-applied (SA, 162/20.00%), and soft-pure (SP, 189/24.00%).

Biglan's classification scheme of academic disciplines (hard (H) vs. soft (S) disciplines and pure (P) vs. applied (A) disciplines) has been credited as the most cited organizational system of academic disciplines in tertiary education [68–70]. Many studies have also provided evidence supporting the validity of this classification [69]. Over the years, research has indicated that Biglan's typology is correlated with differences in many other properties and serves as an appropriate mechanism to organize discipline-specific knowledge or epistemologies [38] and design and deliver courses for students with different learning style preferences [41]. Therefore, this classification provides a convenient framework to explore differences across

disciplinary boundaries. In general, HA disciplines include engineering, HP disciplines include the so-called natural sciences, SA disciplines include the social sciences, and SP disciplines include the humanities [41, 68, 71].

## Instrument

In learning style research, it is difficult to select an instrument to measure the subjects' learning styles [72]. The criteria used for the selection of a learning style instrument in this study include the following: 1) successful use of the instrument in previous studies, 2) demonstrated validity and reliability, 3) a match between the purpose of the instrument and the aim of this study and 4) open access to the questionnaire.

The Felder and Soloman's ILS questionnaire, which was built based on the FSLSM, was adopted in the present study to investigate students' learning styles across different disciplines. First, the FSLSM is recognized as the most commonly used model for measuring individual learning styles on a general scale [73] in higher education [74] and has remained popular for many years across different disciplines in university settings and beyond. In the age of personalized instruction, this model has breathed new life into areas such as blended learning [75], online distance learning [76], courseware design [56], and intelligent tutoring systems [77, 78]. Second, the FSLSM is based on previous learning style models; the FSLSM integrates all their advantages and is, thus, more comprehensive in delineating students' learning styles [79, 80]. Third, the FSLSM has a good predictive ability with independent testing sets (i.e., unknown learning style objects) [17], which has been repeatedly proven to be a more accurate, reliable, and valid model than most other models for predicting students' learning performance [10, 80]. Fourth, the ILS is a free instrument that can be openly accessed online (URL: https://www.webtools.ncsu.edu/learningstyles/) and has been widely used in the research context [81, 82].

The modified and translated version of the original ILS questionnaire includes 44 questions in total, and 11 questions correspond to each dimension of the Felder-Silverman model as follows: questions 1–11 correspond to dimension 1 (active vs. reflective), questions 12–22 correspond to dimension 2 (sensing vs. intuitive), questions 23–33 correspond to dimension 3 (visual vs. verbal), and questions correspond 34–44 to dimension 4 (sequential vs. global). Each question is followed by five choices on a five-point Likert scale ranging from "strongly agree with A (1)", "agree with A (2)", "neutral (3)", "agree with B (4)" and "strongly agree with B (5)". Option A and option B represent the two choices offered in the original ILS questionnaire.

## Ethics statements

The free questionnaires were administered in a single session by specialized staff who collaborated on the investigation. The participants completed all questionnaires individually. The study procedures were in accordance with the ethical standards of the Helsinki Declaration and were approved by the Ethics Committee of the School of International Studies, Zhejiang University. All participants signed written informed consent to authorize their participation in this research. After completion of the informed consent form, each participant was provided a gift (a pen) in gratitude for their contribution and participation.

## Data collection procedure

Before the questionnaires were distributed, the researchers involved in this study contacted faculty members from various departments and requested their help. After permission was given, the printed questionnaires were administered to students under the supervision of their teachers at the end of their English language course. The students were informed of the

purpose and importance of the study and asked to carefully complete the questionnaires. The students were also assured that their personal information would be used for research purposes only. All students provided written informed consent (see S2 File). After the questionnaires were completed and returned, they were thoroughly examined by the researchers such that problematic questionnaires could be identified and excluded from further analysis. All questionnaires eligible for the data analysis had to meet the following two standards: first, all questions must be answered, and second, the answered questions must reflect a reasonable logic. Regarding the few missing values, the median number of a given individual's responses on 11 questions per dimension included in the ILS questionnaire was used to fill the void in each case. In statistics, using the median number to impute missing values is common and acceptable because missing values represent only a small minority of the entire dataset and are assumed to not have a large impact on the final results [83, 84].

In total, 850 questionnaires were administered to the students, and 823 of these questionnaires were retrieved. Of the retrieved questionnaires, the remaining 790 questionnaires were identified as appropriate for further use. After data screening, these questionnaires were organized, and their respective results were translated into an Excel format.

## Data analysis method

During the data analysis, as a library of the SVM, the free package LIBSVM (https://www.csie.ntu.edu.tw/~cjlin/libsvm/) was first applied as an alternative method of data analysis. Then, a traditional ANOVA was performed to examine whether there was a main effect of academic discipline on Chinese students' learning styles. ANOVA could be performed using SPSS, a strong data analysis software that supports a series of statistical analyses. In regard to the examination of the effect of a single or few independent variables, SPSS ANOVA can produce satisfactory results. However, SVM, a classic data mining algorithm, outperforms ANOVA for dataset in which a large number of variables with multidimensions are intertwined and their combined/collective effects influence the classification results. In this study, the research objective was to efficiently differentiate and detect the key features among the 44 factors. Alone, a single factor or few factors might not be significant enough to discriminate the learning styles among the different disciplines. Selected by the SVM, the effects of multiple features may collectively enhance the classification performance. Therefore, the reason for selecting SVM over ANOVA is that in the latter case, the responses on all questions in a single dimension are summed instead of treated as individual scores; thus, the by-item variation is concealed. In addition, the SVM is especially suitable for statistical analysis with high-dimensional factors (usually > 10; 44-dimensional factors were included in this study) and can detect the effects collectively imposed by a feature set [85].

Originally proposed in 1992 [86], the SVM is a supervised learning model related to machine learning algorithms that can be used for classification, data analysis, pattern recognition, and regression analysis. The SVM is an efficient classification model that optimally divides data into two categories and is ranked among the top methods in statistical theory due to its originality and practicality [85]. Due to its robustness, accurate classification, and prediction performance [87–89], the SVM has high reproducibility [90, 91]. Due to the lack of visualization of the computing process of the SVM, the SVM has been described as a "black box" method [92]; however, future studies in the emerging field of explainable artificial intelligence can help solve this problem and convert this approach to a "glass box" method [67]. This algorithm has proven to have a solid theoretical foundation and excellent empirical application in the social sciences, including education [93] and natural language processing [94]. The mechanism underlying the SVM is also presented in Fig 2.

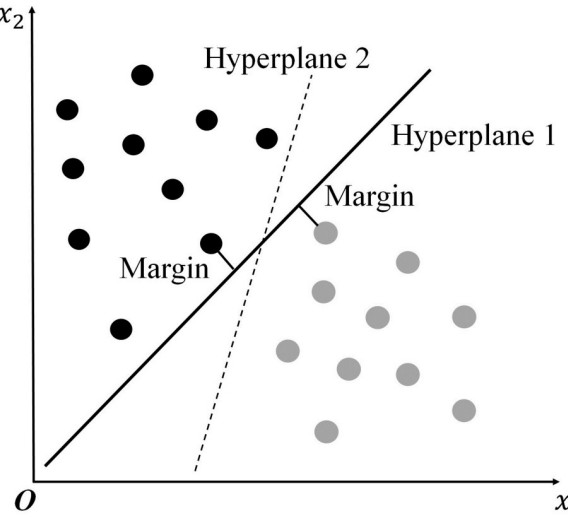

**Fig 2. The mechanism underlying the support vector machine.** Hyperplanes 1 and 2 are two regression lines that divide the data into two groups. Hyperplane 1 is considered the best fitting line because it maximizes the distance between the two groups.

The SVM contains the following two modules: one module is a general-purpose machine learning method, and the other module is a domain-specific kernel function. The SVM training algorithm is used to build a training model that is then used to predict the category to which a new sample instance belongs [95]. When a set of training samples is given, each sample is given the label of one of two categories. To evaluate the performance of SVM models, a confusion matrix, which is a table describing the performance of a classifier on a set of test data for which the true values are known, is used (see Table 1).

Based on the confusion matrix, several indicators were developed to measure the performance of SVM models; of these indicators, the five most common indicators include accuracy (ACC), specificity (SPE), sensitivity (SEN) (also known as 'recall'), area under the receiver operating characteristic curve (AUC), and F-measure. All five values were used in this study as performance evaluators of the SVM models and generally have a value ranging from 0 to 1. The mathematical formulae used to produce these values are provided as follows, along with a brief explanation of their functions:

$$ACC = (TN + TP)/(TP + TN + FP + FN) \tag{1}$$

$$SPE = TN/(TN + FP) \tag{2}$$

**Table 1. Description of a confusion matrix.**

|  | Positive (Predicted) | Negative (Predicted) |
|---|---|---|
| **Positive (Actual)** | True Positive (TP) | False Negative (FN) |
| **Negative (Actual)** | False Positive (FP) | True Negative (TN) |

*Note.* Positive: Observation is positive (e.g., the students belong to this discipline); Negative: Observation is negative (e.g., the students do not belong to this discipline); True Positive (TP): Observation is positive and is predicted to be positive; False Negative (FN): Observation is positive but is predicted to be negative; True Negative (TN): Observation is negative and is predicted to be negative; False Positive (FP): Observation is negative but is predicted to be positive.

$$SEN = TP/(TP + FN) \tag{3}$$

$$AUC = \int_0^1 ROC(t)dt \tag{4}$$

$$F - measure = 2((TP/(TP + FP)) \times SEN)/(TP/(TP + FP) + SEN \tag{5}$$

where

ACC represents the proportion of true results, including both positive and negative results, in the selected population;

SPE represents the proportion of actual negatives that are correctly identified as such;

SEN represents the proportion of actual positives that are correctly identified as such;

AUC is a ranking-based measure of classification performance that can distinguish a randomly chosen positive example from a randomly chosen negative example; and

F-measure is the harmonic mean of precision (another performance indicator) and recall.

The ACC is a good metric frequently applied to indicate the measurement of classification performance, but the combination of the SPE, SEN, AUC, F-measure and ACC may be a measure of enhanced performance assessment and was frequently applied in current studies [96]. In particular, the AUC is a good metric frequently applied to validate the measurement of the general performance of models [97]. The advantage of this measure is that it is invariant to relative class distributions and class-specific error costs [98, 99]. Moreover, to some extent, the AUC is statistically consistent and more discriminating than the ACC with balanced and imbalanced real-world data sets [100], which is especially suitable for unequal samples, such as the HA-HP model in this study. After all data preparations were completed, the data used for the comparisons were extracted separately. First, the processed data of the training set were run by using optimized parameters. Second, the constructed model was used to predict the test set, and the five indicators of the fivefold cross-validation and fivefold average were obtained. Cross-validation is a general validation procedure used to assess how well the results of a statistical analysis generalize to an independent data set, which is used to evaluate the stability of the statistical model. K-fold cross-validation is commonly used to search for the best hyperparameters of SVM to achieve the highest accuracy performance [101]. In particular, fivefold, tenfold, and leave-one-out cross-validation are typically used versions of k-fold cross-validation [102, 103]. Fivefold cross-validation was selected because fivefold validation can generally achieve a good prediction performance [103, 104] and has been commonly used as a popular rule of thumb supported by empirical evidence [105]. In this study, five folds (groups) of subsets were randomly divided from the entire set by the SVM, and four folds (training sample) of these subsets were randomly selected to develop a prediction model, while the remaining one fold (test sample) was used for validation. The above functions were all implemented with Python Programming Language version 3.7.0 (URL: https://www.python.org/).

Then, SVM-RFE, which is an embedded feature selection strategy that was first applied to identify differentially expressed genes between patients and healthy individuals [106], was adopted. SVM-RFE has proven to be more robust to data overfitting than other feature selection techniques and has shown its power in many fields [107]. This approach works by removing one feature each time with the smallest weight iteratively to a feature rank until a group of highly weighted features were selected. After this feature selection procedure, several SVM models were again constructed based on these selected features. The performance of the new

models is compared to that of the original models with all features included. The experimental process is provided in Fig 3 for the ease of reference.

## Results

The classification results produced by SVM and the ranking of the top 20 features produced by SVM-RFE were listed in Table 2. Twenty variables have been selected in this study for two reasons: a data-based reason and a literature-based reason. First, it is clear that models composed of 20 features generally have a better performance than the original models. The performance of models with more than 20 is negatively influenced. Second, SVM-based studies in the social sciences have identified 20 to 30 features as a good number for an optimal feature set [108], and 20 features were selected for inclusion in the optimal feature set [95]. Therefore, in this study, the top 20 features were selected for subsequent analysis, as proposed in previous analyses that yielded accepted measurement rates. These 20 features retained most of the useful information from all 44 factors but with fewer feature numbers, which showed satisfactory representation [96].

### Results of RQ (1) What are the differentiability levels among different academic disciplines based on students' learning styles?

To further measure the performance of the differentiability among students' disciplines, the collected data were examined with the SVM algorithm. As shown in Table 2, the five performance indicators, namely, the ACC, SPE, SEN, AUC and F-measure, were utilized to measure the SVM models. Regarding the two general performance indicators, i.e., the ACC value and AUC value, the HA-HP, HA-SA, and HA-SP-based models yielded a classification capacity of approximately 70.00%, indicating that the students in these disciplines showed a relatively large difference. In contrast, the models based on the H-S, A-P, HP-SA, HP-SP, and SA-SP disciplines only showed a moderate classification capacity (above 55.00%). This finding suggests

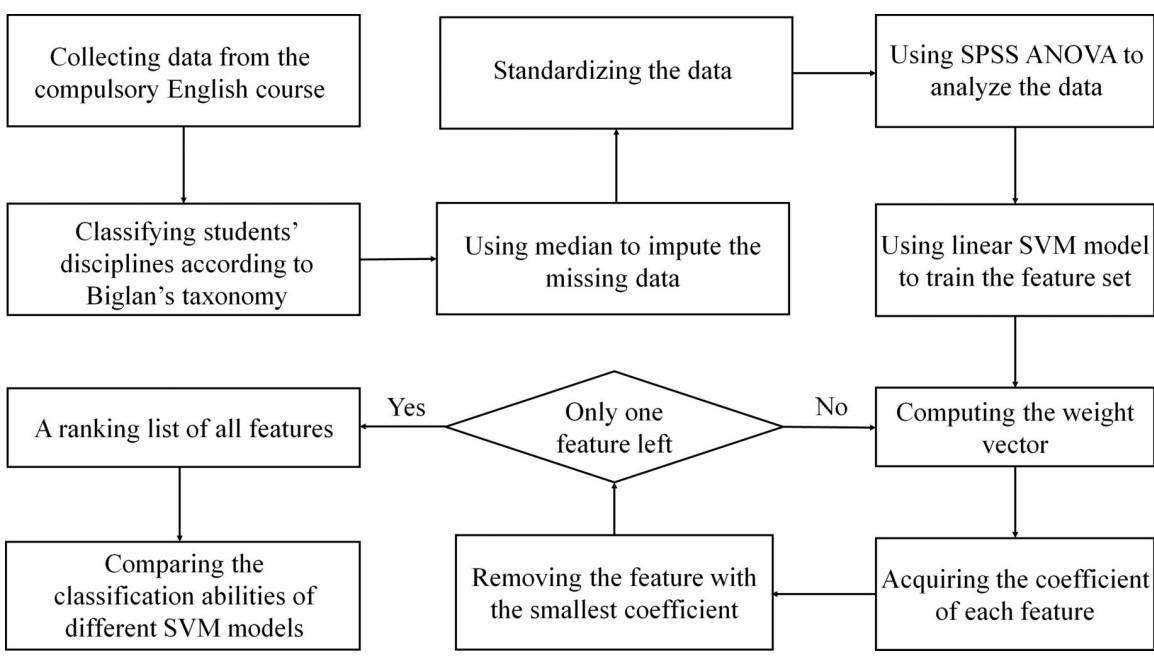

**Fig 3. Experimental process and working mechanism of SVM and SVM-RFE.**

**Table 2. Results produced by SVM and SVM-RFE.**

| Model | Algorithm | ACC | SPE | SEN | AUC | F-measure | Top 20 features |
|-------|-----------|-----|-----|-----|-----|-----------|-----------------|
| H-S | SVM | 66.67% | 69.32% | 63.38% | 66.35% | 66.67% | 1, 2, 4, 5, 7, 8, 9, 10, 11, 14, 21, 22, 23, 26, 27, 29, 31, 32, 35, 39 |
| | SVM-RFE | 87.30% | 88.18% | 86.20% | 87.19% | 87.30% | |
| A-P | SVM | 64.52% | 77.78% | 55.08% | 64.93% | 64.52% | 3, 4, 5, 6, 7, 11, 15, 17, 18, 19, 21, 30, 31, 32, 33, 35, 39, 42, 43, 44 |
| | SVM-RFE | 73.44% | 86.55% | 79.28% | 75.09% | 74.44% | |
| HA-HP | SVM | 69.32% | 80.00% | 67.95% | 73.97% | 69.32% | 1, 4, 5, 6, 7, 8, 10, 12, 13, 14, 19, 21, 26, 28, 29, 31, 34, 39, 40, 44 |
| | SVM-RFE | 82.59% | 82.30% | 78.26% | 89.13% | 79.59% | |
| HA-SA | SVM | 68.47% | 63.64% | 70.51% | 67.07% | 68.47% | 1, 2, 4, 7, 8, 9, 10, 11, 14, 18, 21, 26, 27, 29, 32, 35, 37, 38, 39, 41 |
| | SVM-RFE | 77.74% | 75.00% | 68.22% | 78.11% | 77.74% | |
| HA-SP | SVM | 68.97% | 73.68% | 66.67% | 70.18% | 68.97% | 1, 2, 7, 9, 10, 11, 12, 14, 19, 20, 21, 22, 23, 27, 31, 32, 34, 35, 37, 39 |
| | SVM-RFE | 76.09% | 74.00% | 67.70% | 77.35% | 69.09% | |
| HP-SA | SVM | 55.16% | 56.52% | 58.00% | 57.76% | 56.16% | 1, 2, 4, 5, 6, 7, 9, 10, 12, 15, 16, 21, 26, 27, 28, 29, 31, 33, 39, 44 |
| | SVM-RFE | 74.63% | 72.00% | 70.04% | 75.02% | 69.63% | |
| HP-SP | SVM | 58.33% | 57.90% | 60.00% | 58.95% | 58.33% | 1, 2, 5, 7, 8, 12, 13, 15, 16, 18, 19, 21, 23, 27, 29, 32, 33, 34, 35, 40 |
| | SVM-RFE | 70.83% | 75.00% | 68.00% | 72.50% | 70.83% | |
| SA-SP | SVM | 60.00% | 63.16% | 56.25% | 59.70% | 60.00% | 3, 4, 5, 7, 8, 9, 11, 13, 15, 18, 19, 21, 27, 31, 32, 35, 37, 39, 41, 44 |
| | SVM-RFE | 70.46% | 75.00% | 76.25% | 71.28% | 71.46% | |

*Note*: Indicators in the upper row belong to original SVM models, while those in the lower row belong to the models with 20 features.

that these five SVM models were not as effective as the other three models in differentiating students among these disciplines based on their learning styles. The highest ACC and AUC values were obtained in the model based on the HA-HP disciplines, while the lowest values were obtained in the model based on the HP-SA disciplines. As shown in Table 2, the AUCs of the different models ranged from 57.76% (HP-SA) to 73.97% (HA-HP).

To compare the results of the SVM model with another statistical analysis, an ANOVA was applied. Prior to the main analysis, the students' responses in each ILS dimension were summed to obtain a composite score. All assumptions of ANOVA were checked, and no serious violations were observed. Then, an ANOVA was performed with academic discipline as the independent variable and the students' learning styles as the dependent variable. The results of the ANOVA showed that there was no statistically significant difference in the group means of the students' learning styles in Dimension 1, $F(3, 786) = 2.56$, $p = .054$, Dimension 2, $F(3, 786) = 0.422$, $p = .74$, or Dimension 3, $F(3, 786) = 0.90$, $p = .443$. However, in Dimension 4, a statistically significant difference was found in the group means of the students' learning styles, $F(3, 786) = 0.90$, $p = .005$. As the samples in the four groups were unbalanced, post hoc comparisons using Scheffé's method were performed, demonstrating that the means of the students' learning styles significantly differed only between the HA ($M = 31.04$, $SD = 4.986$) and SP ($M = 29.55$, $SD = 5.492$) disciplines, 95.00% CI for MD [0.19, 2.78], $p = .016$, whereas the other disciplinary models showed no significant differences. When compared with the results obtained from the SVM models, the three models (HA-HP, HA-SA, and HA-SP models) presented satisfactory differentiability capability of approximately 70.00% based on the five indicators.

In the case of a significant result, it was difficult to determine which questions were representative of the significant difference. With a nonsignificant result, it was possible that certain questions might be relevant in differentiating the participants. However, this problem was circumvented in the SVM, where each individual question was treated as a variable and a value was assigned to indicate its relative importance in the questionnaire. Using SVM also

circumvented the inherent problems with traditional significance testing, especially the reliance on p-values, which might become biased in the case of multiple comparisons [109].

## Results of RQ (2) What are the key features that can be selected to determine the collective impact on differentiation by a machine learning algorithm?

To examine whether the model performance improved as a result of this feature selection procedure, the 20 selected features were submitted to another round of SVM analysis. The same five performance indicators were used to measure the model performance (see Table 2). By comparing the performance of the SVM model and that of the SVM-RFE model presented in Table 2, except for the HA-SP model, all other models presented a similar or improved performance after the feature selection process. In particular, the improvement in the HA-HP and HP-SA models was quite remarkable. For instance, in the HA-HP model, the ACC value increased from 69.32% in the SVM model to 82.59% in the SVM-RFE model, and the AUC score substantially increased from 73.97% in the SVM model to 89.13% in the SVM-RFE model. This finding suggests that the feature selection process refined the model's classification accuracy and that the 20 features selected, out of all 44 factors, carry substantive information that might be informative for exploring disciplinary differences. Although results for the indicators of the 20 selected features were not very high, all five indicators above 65.00% showed that the model was still representative because only 20 of 44 factors could present the classification capability. Considering that there was a significant reduction in the number of questions used for the model construction in SVM-RFE (compared with those used for the SVM model), the newly identified top 20 features by SVM-RFE were effective enough to preserve the differential ability of all 44 questions. Thus, these newly identified top 20 factors could be recognized as key differential features for distinguishing two distinct disciplines.

To identify these top 20 features in eight models (see Table 2), SVM-RFE was applied to rank order all 44 features contained in the ILS questionnaire. To facilitate a detailed understanding of what these features represent, the questions related to the top 20 features in the HA-HP model are listed in Table 3 for ease of reference.

## Results of RQ (3) What are the collective impacts of optimal feature sets?

The collective impacts of optimal feature sets could be interpreted from four aspects, namely, the complexities of students' learning styles, the appropriate choice of SVM, the ranking of SVM-RFE and multiple detailed comparisons between students from different disciplines. First, the FSLSM considers the fact that students' learning styles are shaped by a series of factors during the growth process, which intertwine and interact with each other. Considering the complex dynamics of the learning style, selecting an approach that could detect the combined effects of a group of variables is needed. Second, recent years have witnessed the emergence of data mining approaches to explore students learning styles [28, 48–50, 110]. Specifically, as one of the top machine learning algorithms, the SVM excels in identifying the combined effects of high-order factors [87]. In this study, the SVM has proven to perform well in classifying students' learning styles across different disciplines, with every indicator being acceptable. Third, the combination of SVM with RFE could enable the simultaneous discovery of multiple features that collectively determine classification. Notably, although SVM-FRE could rank the importance of the features, they should be regarded as an entire optimal feature set. In other words, the combination of these 20 features, rather than a single factor, could differentiate students' learning styles across different academic disciplines. Last but not least, the multiple comparisons between different SVM models of discipline provide the most effective

**Table 3. Question descriptions of the top 20 features in the HA-HP model.**

| Question Number | Question | Answer Option |
|---|---|---|
| 1 | I understand something better after I | A. try it out. |
| | | B. think it through. |
| 4 | I tend to | A. understand the details of a subject but may be fuzzy about its overall structure. |
| | | B. understand the overall structure but may be fuzzy about the details. |
| 5 | When I am learning something new, it helps me to | A. talk about it. |
| | | B. think about it. |
| 6 | If I were a teacher, I would rather teach a course | A. that addresses facts and real-life situations. |
| | | B. that addresses ideas and theories. |
| 7 | I prefer to obtain new information from | A. pictures, diagrams, graphs, or maps. |
| | | B. written directions or verbal information. |
| 8 | Once I understand | A. all the parts, I understand the whole thing. |
| | | B. the whole thing, I see how the parts fit. |
| 10 | I find it easier | A. to learn facts. |
| | | B. to learn concepts. |
| 12 | When I solve math problems | A. I usually work my way to the solutions one step at a time. |
| | | B. I often just see the solutions but then have to struggle to figure out the steps to get to them. |
| 13 | In the classes I have taken | A. I usually got to know many students. |
| | | B. I rarely got to know many students. |
| 14 | In reading nonfiction, I prefer | A. something that teaches me new facts or tells me how to do something. |
| | | B. something that gives me new ideas to think about. |
| 19 | I remember best | A. what I see. |
| | | B. what I hear. |
| 21 | I prefer to study | A. in a study group. |
| | | B. alone. |
| 26 | When I am reading for enjoyment, I like writers to | A. clearly say what they mean. |
| | | B. say things in creative, interesting ways. |
| 28 | When considering a body of information, I am more likely to | A. focus on the details and miss the big picture. |
| | | B. try to understand the big picture before getting into the details. |
| 29 | I more easily remember | A. something I have done. |
| | | B. something I have thought a lot about. |
| 31 | When someone is showing me data, I prefer | A. charts or graphs. |
| | | B. text summarizing the results. |
| 34 | I consider it higher praise to call someone | A. sensible. |
| | | B. imaginative. |
| 39 | For entertainment, I would rather | A. watch television. |
| | | B. read a book. |
| 40 | Some teachers start their lectures with an outline of what they will cover. Such outlines are | A. somewhat helpful to me. |
| | | B. very helpful to me. |
| 44 | When solving problems in a group, I would be more likely to | A. think of the steps in the solution process. |
| | | B. think of possible consequences or applications of the solution in a wide range of areas. |

*Note*. Question descriptions and answer options were openly accessed online from the ILS (URL: https://www.webtools.ncsu.edu/learningstyles/).

learning style factors, giving researchers clues to the nuanced differences between students' learning styles. It can be seen that students from different academic disciplines understand, see

and reflect things from individualized perspectives. The 20 most effective factors for all models scattered within 1 to 44, verifying students' different learning styles in 4 dimensions. Therefore, the FSLSM provides a useful and effective tool for evaluating students' learning styles from a rather comprehensive point of view.

## Discussion

The following discussions address the three research questions explored in the current study.

### Levels of differentiability among various academic disciplines based on students' learning styles with SVM

The results suggest that SVM is an effective approach for classification in the blended learning context in which students with diverse disciplinary backgrounds can be distinguished from each other according to their learning styles. All performance indicators presented in Tables 2 and 3 remain above the baseline of 50.00%, suggesting that between each two disciplines, students' learning style differences can be identified. To some extent, these differences can be identified with a relatively satisfactory classification capability (e.g., 69.32% of the ACC and 73.97% of the AUC in the HA-HP model shown in Table 2). Further support for the SVM algorithm is obtained from the SVM-RFE constructed to assess the rank of the factors' classification capacity, and all values also remained above the baseline value, while some values reached a relatively high classification capability (e.g., 82.59% of the ACC and 89.13% of the AUC in the HA-HP model shown in Table 2). While the results obtained mostly show a moderate ACC and AUC, they still provide some validity evidence supporting the role of SVM as an effective binary classifier in the educational context. However, while these differences are noteworthy, the similarities among students in different disciplines also deserve attention. The results reported above indicate that in some disciplines, the classification capacity is not relatively high; this was the case for the model based on the SA-SP disciplines.

Regarding low differentiability, one explanation might be the indistinct classification of some emerging "soft disciplines." It was noted that psychology, for example, could be identified as "a discipline that can be considered predominantly 'soft' and slightly 'purer' than 'applied' in nature" [111] (p. 43–53), which could have blurred the line between the SA and SP disciplines. As there is now no impassable gulf separating the SA and SP disciplines, their disciplinary differences may have diminished in the common practice of lecturing in classrooms. Another reason comes from the different cultivation models of "soft disciplines" and "hard disciplines" for sample students. In their high school, sample students are generally divided into liberal art students and science students and are then trained in different environments of knowledge impartation. The two-year unrelenting and intensive training makes it possible for liberal art students to develop a similar thinking and cognitive pattern that is persistent. After the college entrance examination, most liberal art students select SA or SP majors. However, a year or more of study in university does not exert strong effects on their learning styles, which explains why a multitude of researchers have traditionally investigated the SA and SP disciplines together, calling them simply "social science" or "soft disciplines" compared with "natural science" or "hard disciplines". There have been numerous contributions pointing out similarities in the learning styles of students from "soft disciplines" [37, 112–114]. However, students majoring in natural science exhibit considerable differences in learning styles, demonstrating that the talent cultivation model of "hard disciplines" in universities is to some extent more influential on students' learning styles than that of the "soft disciplines". Further compelling interpretations of this phenomenon await only the development of a sufficient level of accumulated knowledge among scholars in this area.

In general, these results are consistent with those reported in many previous studies based on the Felder-Silverman model. These studies tested the precision of different computational approaches in identifying and differentiating the learning styles of students. For example, by means of a Bayesian network (BN), an investigation obtained an overall precision of 58.00% in the active/reflective dimension, 77.00% in the sensing/intuitive dimension and 63.00% in the sequential/global dimension (the visual/verbal dimension was not considered) [81]. With the help of the keyword attributes of learning objects selected by students, a precision of 70.00% in the active/reflective dimension, 73.30% in the sensing/intuitive dimension, 73.30% in the sequential/global dimension and 53.30% in the visual/verbal dimension was obtained [115].

These results add to a growing body of evidence expanding the scope of the application of the SVM algorithm. Currently, the applications of the SVM algorithm still reside largely in engineering or other hard disciplines despite some tentative trials in the humanities and social sciences [26]. In addition, as cross-disciplines increase in current higher education, it is essential to match the tailored learning styles of students and researchers studying interdisciplinary subjects, such as the HA, HP, SA and SP disciplines. Therefore, the current study is the first to incorporate such a machine learning algorithm into interdisciplinary blended learning and has broader relevance to further learning style-related theoretical or empirical investigations.

## Verification of the features included in the optimal feature sets

Features included in the optimal feature sets provided mixed findings compared with previous studies. Some of the 20 identified features are verified and consistent with previous studies. A close examination of the individual questions included in the feature sets can offer some useful insights into the underlying psychological processes. For example, in six of the eight models constructed, Question 1 ("I understand something better after I try it out/think it through") appears as the feature with the number 1 ranking, highlighting the great importance attached to this question. This question mainly reflects the dichotomy between experimentation and introspection. A possible revelation is that students across disciplines dramatically differ in how they process tasks, with the possible exception of the SA-SP disciplines. This difference has been supported by many previous studies. For example, it was found that technical students tended to be more tactile than those in the social sciences [116], and engineering students (known as HA in this study) were more inclined toward concrete and pragmatic learning styles [117]. Similarly, it was explored that engineering students prefer "a logical learning style over visual, verbal, aural, physical or solitary learning styles" [37] (p. 122), while social sciences (known as SA in this study) students prefer a social learning style to a logical learning style. Although these studies differ in their focus to a certain degree, they provide an approximate idea of the potential differences among students in their relative disciplines. In general, students in the applied disciplines show a tendency to experiment with tasks, while those in the pure disciplines are more inclined towards introspective practices, such as an obsession with theories. For instance, in Biglan's taxonomy of academic disciplines, students in HP disciplines prefer abstract rules and theories, while students in SA disciplines favor application [67]. Additionally, Question 10 ("I find it easier to learn facts/to learn concepts") is similar to Question 1, as both questions indicate a certain level of abstraction or concreteness. The difference between facts and concepts is closely related to the classification difference between declarative knowledge and procedural knowledge in cognitive psychology [35, 38]. Declarative knowledge is static and similar to facts, while procedural knowledge is more dynamic and primarily concerned with operational steps. Students' preferences for facts or concepts closely correspond to this psychological distinction.

In addition, Questions 2, 4, 7, and 9 also occur frequently in the 20 features selected for the different models. Question 2 ("I would rather be considered realistic/innovative") concerns taking chances. This question reflects a difference in perspective, i.e., whether the focus should be on obtaining pragmatic results or seeking original solutions. This difference cannot be easily connected to the disciplinary factor. Instead, there are numerous factors, e.g., genetic, social and psychological factors, that may play a strong role in defining this trait. The academic discipline only serves to strengthen or diminish this difference. For instance, decades of research in psychology have shown that males are more inclined towards risk taking than females [118–121]. A careful examination of the current academic landscape reveals a gender difference; more females choose soft disciplines than males, and more males choose hard disciplines than females. This situation builds a disciplinary wall classifying students into specific categories, potentially strengthening the disciplinary effect. For example, Question 9 ("In a study group working on difficult material, I am more likely to jump in and contribute ideas/sit back and listen") emphasizes the distinction between active participation and introspective thinking, reflecting an underlying psychological propensity in blended learning. Within this context, the significance of this question could also be explained by the psychological evaluation of "loss and gain", as students' different learning styles are associated with expected reward values and their internal motivational drives, which are determined by their personality traits [122]. When faced with the risk of "losing face", whether students will express their ideas in front of a group of people depends largely on their risk and stress management capabilities and the presence of an appropriate motivation system.

The other two questions also convey similar messages regarding personality differences. Question 4 concerns how individuals perceive the world, while Question 7 concerns the preferred modality of information processing. Evidence of disciplinary differences in these respects was also reported [35, 123–125]. The other questions, such as Questions 21, 27, and 39, show different aspects of potential personality differences and are mostly consistent with the previous discussion. This might also be a vivid reflection of the multi-faceted effects of blended learning, which may differ in their consonance with the features of each discipline. First, teachers from different domains use technology in different ways, and student from different disciplines may view blended learning differently. For instance, the characteristics of soft-applied fields entail specialized customization in blended courses, further broadening the gulf between different subjects [126]. Second, although blended learning is generally recognized as a stimulus to students' innovation [127], some students who are used to an instructivist approach in which the educator acts as a 'sage on the stage' will find it difficult to adapt to a social constructivist approach in which the educator serves as a 'guide on the side' [128]. This difficulty might not only negatively affect students' academic performance but also latently magnify the effects of different academic disciplines.

## Interpretation of the collective impact of optimal feature sets

In each SVM model based on a two-discipline model, the 20 key features (collectively known as an optimal feature set) selected exert a concerted effect on students' learning styles across different disciplines (see Table 2). A broad examination of the distribution of collective impact of each feature set with 20 features in the eight discipline models suggests that it is especially imperative considering the emerging cross-disciplines in academia. Current higher education often involves courses with crossed disciplines and students with diverse disciplinary backgrounds. In addition, with the rise of technology-enhanced learning, the design of personalized tutoring systems requires more nuanced information related to student attributes to provide greater adaptability [59]. By identifying these optimal feature sets, such information

becomes accessible. Therefore, understanding such interdisciplinary factors and designing tailor-made instructions are essential for promoting learning success [9]. For example, in an English language classroom in which the students are a blend of HP and SP disciplines, instructors might consider integrating a guiding framework at the beginning of the course and stepwise guidelines during the process such that the needs of both groups are met. With the knowledge that visual style is dominant across disciplines, instructors might include more graphic presentations (e.g., Question 11) in language classrooms rather than continue to use slides or boards filled with words. Furthermore, to achieve effective communication with students and deliver effective teaching, instructors may target these students' combined learning styles. While some methods are already practiced in real life, this study acts as a further reminder of the rationale underlying these practices and thus increases the confidence of both learners and teachers regarding these practices. Therefore, the practical implications of this study mainly concern classroom teachers and educational researchers, who may draw some inspiration for interdisciplinary curriculum design and the tailored application of learning styles to the instructional process.

## Conclusions

This study investigated learning style differences among students with diverse disciplinary backgrounds in a blended English language course based on the Felder-Silverman model. By introducing a novel machine learning algorithm, namely, SVM, for the data analysis, the following conclusions can be reached. First, the multiple performance indicators used in this study confirm that it is feasible to apply learning styles to differentiate various disciplines in students' blended learning processes. These disciplinary differences impact how students engage in their blended learning activities and affect students' ultimate blended learning success. Second, some questions in the ILS questionnaire carry more substantive information about students' learning styles than other questions, and certain underlying psychological processes can be derived. These psychological processes reflect students' discipline-specific epistemologies and represent the possible interaction between the disciplinary background and learning style. In addition, the introduction of SVM in this study can provide inspiration for future studies of a similar type along with the theoretical significance of the above findings.

Despite the notable findings of this study, it is subject to some limitations that may be perfected in further research. First, the current analysis examined the learning styles without allowing for the effects of other personal or contextual factors. The educational productivity model proposed by Walberg underlines the significance of the collected influence of contextual factors on individuals' learning [129]. For example, teachers from different backgrounds and academic disciplines are inclined to select various teaching methods and to create divergent learning environments [130], which should also be investigated thoroughly. The next step is therefore to take into account the effects of educational background, experience, personality and learning experience to gain a more comprehensive understanding of students' learning process in the blended setting.

In conclusion, the findings of this research validate previous findings and offer new perspectives on students' learning styles in a blended learning environment, which provides future implications for educational researchers, policy makers and educational practitioners (i.e., teachers and students). For educational researchers, this study not only highlights the merits of using machine learning algorithms to explore students' learning styles but also provides valuable information on the delicate interactions between blended learning, academic disciplines and learning styles. For policy makers, this analysis provides evidence for a more inclusive but personalized educational policy. For instance, in addition to learning styles, the

linkage among students' education in different phases should be considered. For educational practitioners, this study plays a positive role in promoting student-centered and tailor-made teaching. The findings of this study can help learners of different disciplines develop a more profound understanding of their blended learning tendencies and assist teachers in determining how to bring students' learning styles into full play pedagogically, especially in interdisciplinary courses [131–134].

## Supporting information

**S1 File.**
(DOCX)

**S2 File. Informed consent for participants.**
(DOCX)

**S1 Dataset.**
(XLSX)

## Acknowledgments

The authors would like to thank the anonymous reviewers for their constructive comments on this paper and Miss Ying Zhou for her suggestions during the revision on this paper.

## Author Contributions

**Conceptualization:** Jie Hu.

**Formal analysis:** Jie Hu, Yi Peng, Xueliang Chen.

**Funding acquisition:** Jie Hu.

**Methodology:** Jie Hu.

**Project administration:** Yi Peng.

**Supervision:** Jie Hu.

**Writing – original draft:** Jie Hu, Xueliang Chen.

**Writing – review & editing:** Jie Hu, Yi Peng, Hangyan Yu.

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

           1901667

