## [Decision Letter · Decision Letter 0]

5 Aug 2020

PONE-D-20-14494

Differentiating the learning styles of EFL students with different disciplines in a blended learning setting: Insights from a machine learning-based algorithm

PLOS ONE

Dear Dr. Hu,

Thank you for submitting your manuscript to PLOS ONE. After careful consideration, we feel that it has merit but does not fully meet PLOS ONE’s publication criteria as it currently stands. Therefore, we invite you to submit a revised version of the manuscript that addresses the points raised during the review process.

We look forward to receiving your revised manuscript.

Kind regards,

Haoran Xie

Academic Editor

PLOS ONE

Journal Requirements:

2. Please upload a copy of the questionnaire used in the study as a supplemental file or provide a URL where it can be accessed. Additionally, although you state that faculty gave permission for their students to complete the questionnaire, please state in both your Ethics Statement and your Methods section whether informed consent was given by the participants who completed the questionnaire or whether the need for consent was waived by your ethics approval board. Please state what type of consent was given (i.e., written, verbal, etc.).

3. Please ensure that you include a title page within your main document. We do appreciate that you have a title page document uploaded as a separate file, however, as per our author guidelines (http://journals.plos.org/plosone/s/submission-guidelines#loc-title-page) we do require this to be part of the manuscript file itself and not uploaded separately.

Reviewers' comments:

Reviewer's Responses to Questions

**Comments to the Author**

1. Is the manuscript technically sound, and do the data support the conclusions?

Reviewer #1: Yes

Reviewer #2: No

2. Has the statistical analysis been performed appropriately and rigorously? 

Reviewer #1: Yes

Reviewer #2: Yes

3. Have the authors made all data underlying the findings in their manuscript fully available?

Reviewer #1: Yes

Reviewer #2: Yes

4. Is the manuscript presented in an intelligible fashion and written in standard English?

Reviewer #1: Yes

Reviewer #2: Yes

5. Review Comments to the Author

Reviewer #1: This study aims to introduce an innovative machine learning algorithm known as support vector machine (SVM) and specify the impact of students’ disciplinary background on their learning styles in EFL blended learning, which is interesting and investigation worthy. The method is inspiring and well-designed. The authors may further improve this paper in the following points.

1) There is no literature review in this paper, leading to the insufficient establishment of theoretical framework of this paper. Before leading in the research method, a few questions from three main aspects are supposed to be answered based on the literature: (a) “what is the definition of learning style?” “What are the major learning styles of EFL learning/blended learning/EFL blended learning as shown in the literature?” “From which aspects could learning styles influence EFL learning/blended learning/EFL blended learning?” (b) “How could disciplines influence learning styles?” “How could learning methods influence learning styles?” (c) “What are the explicit limitations of SPSS which results in the incapability of it to address the present research problems?” A rigid theoretical framework would also be helpful in the analysis of the research results so as to make the discussion section better-structured and more in-depth.

2) It is suggested to point out the research questions clearly.

3) It is suggested to add a section about the practical implications for future researchers and educators.

4) The paper is well-presented in general, in spite of a few points in need of further improvement. For example, in the 1st paragraph, the statement “Knowing a student’s learning style has many obvious benefits” may not be in an intelligible fashion as expected on a research paper. The authors are suggested to make revisions to the language.

Reviewer #2: This manuscript tested the precision of different computational approaches in identifying and differentiating college sophomore students’ learning styles based on the Felder–Silverman model. Results showed that although no difference was found in participants’ learn style across different disciplinary groups, SVM-RFE machine learning algorithm showed that learning styles can differentiate various disciplines in EFL students’ blended learning processes as confirmed by the multiple performance indicators and further, some questions in the ILS questionnaire carry more substantive information about students’ learning styles than others. Overall, the manuscript is clearly written and the statistics are robust; I have the following concerns regarding the theoretical assumption and data interpretation of the study before its acceptance for publication.

A major concern is that all the statistical comparisons of the current manuscript are based on the basic assumption that disciplinary differences predict learning style differences. Results showed that SVM-REF machine learning algorithm can be applied to show high percentage of correspondence between learning style and disciplinary backgrounds in some comparisons, such as HA-HP groups, whereas almost no such correspondence was found with ANOVA comparisons, which revealed no significant difference in learning styles across different participant groups. established on this, the manuscript argued that the SVM-REF is logically a better method because it is closer to reflect truth. However, the fallacy here is that the “assumed TRUTH” may not be “the truth”. No evidence showed such a correspondence between disciplinary background and learning style of ALL individuals. Learning styles have been shown to be related to a whole multitude of factors, such as individuals’ personality, leaning background, experience, etc. Disciplinary background is only one of them, maybe a minor one out of many. Thus, the “non-significant result” might be true here. While the “significant result” might not be the truth. This logic fallacy brings my biggest concern regarding the research design of the manuscript.

Hidden in the sophisticated descriptions of all statistical methods and algorithms, the manuscript did not provide literature to support such a clear route and corresponding relationship between disciplinary background and learning style. Ref 24 is related but not directly supporting it. Without consideration of all other factors and only entering the single disciplinary factor into the algorithm arbitrarily is not substantiated. Indeed the results did show non-significant(relatively low performance) in some backgrounds. In the discussion the authors attributed this to the fact that these students chose courses from other disciplines and that lecturing in EFL classrooms regardless of the students’ background information diminished their disciplinary differences. (Line 388-390). Since all students are in such a protocol, I am thinking why there are significant results in other discipline comparisons.

My other concern is that although the tile claims “learning styles of EFL students” explicitly, the current data is not in essence related specifically to blended EFL learning. The questionnaire adopted in the current study is tapping into learners’ general learning style. The only thing that is related to EFL is that questionnaires were collected in EFL class time period. Such items as “When I solve math problems …”(P12) in the questionnaire show that is not speicifically EFL learning style.

Minor:

Line 63-65: Ref26 is emphasizing the importance of real world models instead of using machine learning blindly. However, this paragraph here the authors are highting the merits of machine learning. the logic is not consistent here.

Table titles can be shortened. For example, in table 2 such words as “Constructed in This Study” are redundant.

6. PLOS authors have the option to publish the peer review history of their article (what does this mean?). If published, this will include your full peer review and any attached files.

Reviewer #1: No

Reviewer #2: No

---

## [Author Response · Author response to Decision Letter 0]

14 Nov 2020

We would like to express our sincere thanks for reviewing our manuscript entitled “Differentiating the learning styles of EFL students with different disciplines in a blended learning setting: Insights from a machine learning-based algorithm” and providing us an opportunity to make corrections in it. As per your valuable comments, the manuscript has been revised and all the amendments suggested by the reviewers have been incorporated. Due to limited space here, please kindly review the ‘Authors’ Responses to Reviewers’ Comments’ file for point-to-point responses in detail, which is more than 10 pages. Once again, we are thankful to all of the reviewers for promote the quality of our manuscript!

---

## [Decision Letter · Decision Letter 1]

22 Dec 2020

PONE-D-20-14494R1

Differentiating the learning styles of college students with different disciplines in a blended learning setting: Insights from a machine learning-based algorithm

PLOS ONE

Dear Dr. Hu,

Thank you for submitting your manuscript to PLOS ONE. After careful consideration, we feel that it has merit but does not fully meet PLOS ONE’s publication criteria as it currently stands. Therefore, we invite you to submit a revised version of the manuscript that addresses the points raised during the review process.

We look forward to receiving your revised manuscript.

Kind regards,

Haoran Xie

Academic Editor

PLOS ONE

Additional Editor Comments (if provided):

Please revise the manuscript according to review comments.

Reviewers' comments:

Reviewer's Responses to Questions

**Comments to the Author**

1. If the authors have adequately addressed your comments raised in a previous round of review and you feel that this manuscript is now acceptable for publication, you may indicate that here to bypass the “Comments to the Author” section, enter your conflict of interest statement in the “Confidential to Editor” section, and submit your "Accept" recommendation.

Reviewer #1: All comments have been addressed

Reviewer #2: All comments have been addressed

2. Is the manuscript technically sound, and do the data support the conclusions?

Reviewer #1: Partly

Reviewer #2: Yes

3. Has the statistical analysis been performed appropriately and rigorously? 

Reviewer #1: Yes

Reviewer #2: Yes

4. Have the authors made all data underlying the findings in their manuscript fully available?

Reviewer #1: Yes

Reviewer #2: Yes

5. Is the manuscript presented in an intelligible fashion and written in standard English?

Reviewer #1: No

Reviewer #2: Yes

6. Review Comments to the Author

Reviewer #1: The authors have carefully revised this paper and addressed all the comments. Remarkable improvements are spotted.

1. The title is very informative but can be shorter and catchier.

2. The introduction is a little bit mixed with the literature review. Since this paper involves several different topics, I suggest the authors divide the current “Introduction” into several sections so your readers can follow you more easily. I suggest a division as follows: (1) a short section for a catchy introduction briefly specifying the research background, research rationales and research questions; (2) a section for literature review, in which the descriptions of the theories on learning styles and blended learning are organized separately in different sub-sections and catchy sub-titles; (3) a section for research gaps and research questions.

3. The methodology is well written, with clear organization and sufficient justification.

4. The authors may double check the use of punctuation. For example, in line 564, “soft discipline”. to “soft discipline.”

5. The coherence and logics within the paragraphs may be further improved, especially in the discussion and implication sections. For example, in line 581-587, the authors used “However” in two successive sentences, causing logical incoherence.

6. The authors raised many examples in the discussions, which is good. I suggest the authors put more specific explanations for their arguments in the discussions. Examples may be used as evidence to SUPPORT the arguments and explanations, rather than REPLACE them.

Reviewer #2: The revised manuscript has addressed all my concerns. The revised manuscript is improved significantly and should　be a nice contribution to the literature. The abstract is a big heavy in sentence structure, the authors may consider condense it a little bit before publication.

7. PLOS authors have the option to publish the peer review history of their article (what does this mean?). If published, this will include your full peer review and any attached files.

Reviewer #1: No

Reviewer #2: No

---

## [Author Response · Author response to Decision Letter 1]

4 Jan 2021

Responses to Reviewer #1

Comment 1

The authors have carefully revised this paper and addressed all the comments. Remarkable improvements are spotted.

Response to comment 1

Many thanks to the reviewer for this positive comment.

Comment 2

The title is very informative but can be shorter and catchier.

Response to comment 2

Thanks to Reviewer 1 for this recommendation. We have shortened the title “Differentiating the learning styles of college students with different disciplines in a blended learning setting: Insights from a machine learning-based algorithm” to “Differentiating the learning styles of college students with different disciplines in a college English blended learning setting” to make it precise while still informative enough.

Comment 3

The introduction is a little bit mixed with the literature review. Since this paper involves several different topics, I suggest the authors divide the current “Introduction” into several sections so your readers can follow you more easily. I suggest a division as follows: (1) a short section for a catchy introduction briefly specifying the research background, research rationales and research questions; (2) a section for literature review, in which the descriptions of the theories on learning styles and blended learning are organized separately in different sub-sections and catchy sub-titles; (3) a section for research gaps and research questions.

Response to comment 3

Many thanks to the reviewer for this invaluable suggestion. Since the format of PLoS ONE suggests only five Level 1 headings, namely, introduction, materials and methods, results, discussions, and conclusions; thus, we took the effort to divide the Level 1 introduction section into three Level 2 sub-sections, namely, research background, literature review and research gaps and research questions according to the suggestions to make it clearly structured and friendly readable.

(1) In the research background sub-section, we have briefly introduced the research rationale and research significance. Modifications were made in the manuscript as a newly developed section in blue font, which is also provided here as follows for the ease of reference:

Research background

Learning style, as an integral and vital part of students’ learning process, has always been a long-lasting and hot-debated topic in the field of education and pedagogy……

However, knowledge of the learning styles of students from different disciplines in blended learning environments is limited [11-13]. In an effort to address this gap, the present study is designed to achieve two major objectives. First, it intends to gain insight on how disciplinary background impacts students’ learning styles in a blended learning environment based on data collected in a compulsory college English course in which students across 46 disciplines were enrolled. Second, it introduces a novel machine learning method named the support vector machine (SVM) to the field of education to identify an optimal set of factors that can simultaneously differentiate students of different academic disciplines. This research aims to provide a clearer picture of the relationship between disciplinary factors and the identification of students’ learning styles in a blended learning setting. 

(2) In the literature review sub-section, we have divided the original mixed and confusing part into two sub-sections with clear sub-titles, where the theories of learning style and blended learning were examined successively. In the first Level 3 sub-section entitled “Theories of learning styles”, definitions, influencing factors and models of learning styles were investigated. The examination of the conceptual framework of the research, namely, Felder-Silverman learning style model (FSLSM) was also included. In the second Level 3 sub-section named “Learning styles in an online/blended learning environment”, scholarly works with regard to learning styles in both web-based environment and blended learning environment were examined. The revised part is marked in blue font and is presented here as follows: 

Literature Review

Theories of learning styles

The examination of a constellation of definitions, interpretations and constructions pertaining to the term learning style could provide a solid footing for the related studies, through which the understanding of it can be enriched and extended……

Learning styles vary according to a series of factors, including but not limited to age [22], gender [23], personality [2, 24], learning environment [25] and learning experience [26]. In the higher education context, the academic discipline prominently represents an omnipresent variable that influences students’ distinctive learning styles, which echoes a multitude of investigations [27-35]. One notable study explored the learning styles of students from 4 clusters of disciplines in an academic English language course, ……

Different students receive, process, and respond to information with different learning styles. A theoretical model of learning style can be used to categorize people according to their idiosyncratic learning styles.…….

The FSLSM includes learning styles scattered among four dimensions. Visual learners process information best when it is presented as graphs, pictures, etc.,…….

Learning styles in an online/blended learning environment

Individuals’ learning styles reflect their adaptive orientation to learning and are not fixed personality traits. Consequently, learning styles can vary among diverse contexts, and related research in different contexts is vital to understanding learning styles in greater depth. Web-based technologies eliminate barriers of space and time and have become integrated in individuals’ daily lives and learning habits.…...

Considering that academic disciplines and learning environment are both potent predictors of students’ learning styles, many studies have also concentrated on the effects of academic discipline in a blended learning environment.……

(3) In the sub-section for research gaps and research questions, we have briefly summarized the research gaps of the abovementioned scholarly works and elaborated on the two main objectives. This section is now listed below for the reviewer’s reference, and the modifications were highlighted in blue font here and in the revised manuscript:

Research gaps and research questions

Overall, the research seems to be gaining traction, and new perspectives are continually introduced ……. Therefore, one goal of the present study is to fill this gap and shed light on this topic.

Another issue addressed in this study is the selection of an optimal measurement that can effectively identify and differentiate individual learning styles [43]……These findings across different fields point to the reliability of SVM as an effective statistical tool for identification and differentiation analysis.

Therefore, a comprehensive investigation across the four general disciplines in Biglan’s taxonomy using a strong machine learning approach is needed. Given the existence of the research gaps discussed above, this exploratory study seeks to address the following questions:

1) Can students’ learning styles be applied to differentiate various academic disciplines in the blended learning setting? If so, what are the levels of differentiability among different academic disciplines based on students’ learning styles?

2) What are the key features that can be selected to determine the collective impact on differentiation by a machine learning algorithm?

3) What are collective impacts of optimal feature sets?

Comment 4

The methodology is well written, with clear organization and sufficient justification.

Response to comment 4

We would like to thank the reviewer very much for this positive comment.

Comment 5

The authors may double check the use of punctuation. For example, in line 564, “soft discipline”. to “soft discipline.”

Response to comment 5

Thanks to the reviewer for pointing out this problem. Indeed, according to the rules of punctuation, we should put the full stop in front of the quotation marks in Line 570. At the same time, similar problems were checked throughout the manuscript and were revised if necessary.

Comment 6

The coherence and logics within the paragraphs may be further improved, especially in the discussion and implication sections. For example, in line ‪581-587‬, the authors used “However” in two successive sentences, causing logical incoherence.‬‬‬‬‬‬‬‬‬‬‬‬‬

Response to comment 6

Many thanks to the reviewer for this constructive suggestion. We felt very sorry that we did not pay attention to the repeated use of the word “However” in the arguments in the original manuscript, which caused unnecessary misunderstandings. The first “However” in Line 587 was initially conceived to indicate a huge difference between two group of students, namely, students majoring in natural science and their liberal art counterparts. The second use of “However” was indicative of the insufficient knowledge about the causal effects of this interpretation. According to the suggestions, we have made revisions to this argument and tried to make it more applicable. We have kept the first “However” and changed the expression in Line 591. The revised sentence is provided below in blue font for reference:

However, students majoring in natural science exhibit considerable differences in learning styles, demonstrating that the talent cultivation model of “hard disciplines” in universities is to some extent more influential on students’ learning styles than that of the “soft disciplines”. Further compelling interpretations of this phenomenon await only the development of a sufficient level of accumulated knowledge among scholars in this area.

Comment 7

The authors raised many examples in the discussions, which is good. I suggest the authors put more specific explanations for their arguments in the discussions. Examples may be used as evidence to SUPPORT the arguments and explanations, rather than REPLACE them.

Response to comment 7

Our heartfelt thanks go to the reviewer for this invaluable and helpful suggestion. Following the reviewer’s suggestion, we did carefully add more specific explanations of the results, using examples as evidence to support our arguments. The revised part is provided below in blue font for ease of reference:

In addition, Questions 2, 4, 7, and 9 also occur frequently in the 20 features selected for the different models.…… Within this context, the significance of this question could also be explained by the psychological evaluation of “loss and gain”, as students’ different learning styles are associated with expected reward values and their internal motivational drives, which are determined by their personality traits [112]. When faced with the risk of “loosing face”, whether students will express their ideas in front of a group of people depends largely on their risk and stress management capabilities and the presence of an appropriate motivation system.

The other two questions also convey similar messages regarding personality differences.…... This might also be a vivid reflection of the multi-faceted effects of blended learning, which may differ in their consonance with the features of each discipline. First, teachers from different domains use technology in different ways, and student from different disciplines may view blended learning differently. For instance, the characteristics of soft-applied fields entail specialized customization in blended courses, further broadening the gulf between different subjects [116]. Second, although blended learning is generally recognized as a stimulus to students’ innovation [117], some students who are used to an instructivist approach in which the educator acts as a ‘sage on the stage’ will find it difficult to adapt to a social constructivist approach in which the educator serves as a ‘guide on the side’ [118]. This difficulty might not only negatively affect students’ academic performance but also latently magnify the effects of different academic disciplines.

Comment 8

Is the manuscript presented in an intelligible fashion and written in standard English?

Reviewer #1: No

Response to comment 8

Thank you very much for the reviewer’s valuable suggestion. During the revision, we carefully proof-read the entire manuscript and improved the language by asking our colleagues to review the main text and the supporting materials. The native language of our colleagues is English, and they are all from my research fields in my university. Base on their suggestions, necessary modifications and improvements to the manuscript were made, which were all highlighted in blue font in the revised manuscript.

In addition, this version of manuscript was sent to the professional proof-reading agency American Journal Experts (AJE) for English language editing, which is the official English language editing agency partner of PLOS. The proof-reading quality certificate (AJE Order NO.: MCF1YPS3) with the title of this manuscript was offered.

 Responses to Reviewer #2

Comment 1

The revised manuscript has addressed all my concerns. The revised manuscript is improved significantly and should be a nice contribution to the literature.

Response to comment 1

Many thanks to the reviewer for this positive comment.

Comment 2

The abstract is a big heavy in sentence structure, the authors may consider condense it a little bit before publication.

Response to comment 2

The reviewer’s valuable suggestion is greatly appreciated. According to the suggestion, we have condensed the abstract according to the suggestion and made the sentence structure more precise. The revised abstract is provided as below in blue font for the reviewer’s reference:

Learning styles are critical to educational psychology, especially when investigating various contextual factors that interact with individual learning styles. Drawing upon Biglan’s taxonomy of academic tribes, this study systematically analyzed the learning styles of 790 sophomores in a blended learning course with 46 specializations using a novel machine learning algorithm called the support vector machine (SVM). Moreover, an SVM-based recursive feature elimination (SVM-RFE) technique was integrated to identify the differential features among distinct disciplines. The findings of this study shed light on the optimal feature sets that collectively impacted the students’ learning style differences in a college English blended learning setting.

---

## [Decision Letter · Decision Letter 2]

2 Feb 2021

PONE-D-20-14494R2

Differentiating the leanrning styles of college students with different disciplines in a college English blended learning setting

PLOS ONE

Dear Dr. Hu,

Thank you for submitting your manuscript to PLOS ONE. After careful consideration, we feel that it has merit but does not fully meet PLOS ONE’s publication criteria as it currently stands. Therefore, we invite you to submit a revised version of the manuscript that addresses the points raised during the review process.

We look forward to receiving your revised manuscript.

Kind regards,

Haoran Xie

Academic Editor

PLOS ONE

Additional Editor Comments (if provided):

Please revise the paper according to review comments.

Reviewers' comments:

Reviewer's Responses to Questions

**Comments to the Author**

1. If the authors have adequately addressed your comments raised in a previous round of review and you feel that this manuscript is now acceptable for publication, you may indicate that here to bypass the “Comments to the Author” section, enter your conflict of interest statement in the “Confidential to Editor” section, and submit your "Accept" recommendation.

Reviewer #1: (No Response)

Reviewer #2: All comments have been addressed

2. Is the manuscript technically sound, and do the data support the conclusions?

Reviewer #1: Partly

Reviewer #2: Yes

3. Has the statistical analysis been performed appropriately and rigorously? 

Reviewer #1: Yes

Reviewer #2: Yes

4. Have the authors made all data underlying the findings in their manuscript fully available?

Reviewer #1: Yes

Reviewer #2: Yes

5. Is the manuscript presented in an intelligible fashion and written in standard English?

Reviewer #1: No

Reviewer #2: Yes

6. Review Comments to the Author

Reviewer #1: PONE-D-20-14494_R2

Differentiating the learning styles of college students with different disciplines in a college English blended learning setting

Overall, this article was well written and organized. Additionally, great revisions have been made based on the suggestions by previous reviewers, which was made this article more convincing. However, some details should be offered, and minor revisions should be made before publication. They are listed as follows:

1. Though this article has been proofread, some language concerns existed. The authors should carefully revise the language before re-submit the manuscript. (1) Inaccurate wording such as ‘impacted the students’ learning style differences (Abstract)’, ‘..a clearer picture.. (Line 66, what did the author compared with?)’, and ‘an omnipresent’. (2) Wordy sentences, such as ‘.. has always been a long-lasting and hot-debated topic….. (Line 48)’ and ‘the present study is designed to achieve two major objectives (Line 60)’. These two sentences might be re-wrote as ‘… has been constantly discussed’ and ‘this study aims to achieve …’. (3) Hard-to-read long sentences, such as ‘First, …. 46 disciplines were enrolled (Line 60-63)’. The authors are suggested to re-organize the long sentences into a few logically connected short sentences, which are easy for the readers to follow. (5) The authors are suggested to use hedge words instead of the words with a strong tongue (e.g., potent, omnipresent, a constellation of)

2. Literature review. (1) Why did the authors write ‘The examination of a constellation … and extended (Line 71-73)’ as the beginning of this paragraph? What is the connection between this sentence and the following contents? (2) Could the authors add some details about ‘analytical lenses (Line 88)’? (3) It might not be appropriate to use the conjunction word ‘Despite (Line 88)’, because I could not figure out the adversative relation. I guess that the author may intend to state ‘FSLSM was adopted most’. (4) I suggest that the author add a subheading (maybe: Learning style and FSLSM) before the second paragraph in the Literature review section to orient the readers. (5) The transition from the second paragraph to the third paragraph was not smooth; the authors are recommended to add one or two transitional sentences. (6) Any other reasons for the authors’ application of FSLSM? (7) The reviews on learning styles in an online/blended learning environment were not enough, the authors are suggested to add additional studies and summarize the main findings of them. Based on this, the authors could propose research gaps and questions.

3. Materials and methods. This part is organized with sufficient justification, but the authors should add some details. (1) Would it possible to draw a simple flow chart to describe the whole experiment process? (2) For the first time to use ‘ILS’, please write the full name.

4. Results. (1) It is a bit hard to follow the research results described by the authors, would it possible to generally summarize the main results in a table? (2) Why did the authors choose 20 features? (3) The authors are recommended to present the results as per the proposed questions. For example, subheading ‘Results of question one’.

5. Discussion. Try to write with a general-specific structure. Writing a summative sentence at the beginning of each paragraph to present the core idea, then explain it.

6. The authors had better avoid using one-sentence paragraphs in an academic article.

7. The language may be more concise and information-intensive. For example, phrases and sentences like “In the field of learning style research, one great difficulty is the selection of 282 proper instruments to measure the subjects’ learning styles [59].” and “Among the various inventories of learning styles” may be deleted.

8. It is observed that some sections begin with “First.” I suggest the authors add a topic sentence before “First,” briefly illustrating what will be presented. The titles could not replace topic sentences.

9. Based on the in-depth discussions and fruitful results, the authors may propose some implications for future researchers and educators.

Reviewer #2: All my previoius concerns and comments have been addressed satisfactorilly. I have no further comments.

7. PLOS authors have the option to publish the peer review history of their article (what does this mean?). If published, this will include your full peer review and any attached files.

Reviewer #1: No

Reviewer #2: No

---

## [Author Response · Author response to Decision Letter 2]

27 Mar 2021

Dear Esteemed Reviewer 1,

We would like to express our sincere gratitude for reviewing our manuscript entitled “Differentiating the learning styles of college students in different disciplines in a college English blended learning setting” and providing us an opportunity to make corrections in it. We have revised our manuscript according to the valuable comments of the reviewer. The following is an outline of our point-to-point responses and corresponding revisions made to our manuscript.

Once again, we are thankful to you for promoting the quality of our manuscript!

Best regards,

Authors

---

## [Decision Letter · Decision Letter 3]

29 Apr 2021

Differentiating the leanrning styles of college students in different disciplines in a college English blended learning setting

PONE-D-20-14494R3

Dear Dr. Hu,

We’re pleased to inform you that your manuscript has been judged scientifically suitable for publication and will be formally accepted for publication once it meets all outstanding technical requirements.

Kind regards,

Haoran Xie

Academic Editor

PLOS ONE

Additional Editor Comments (optional):

Reviewers' comments:

Reviewer's Responses to Questions

**Comments to the Author**

1. If the authors have adequately addressed your comments raised in a previous round of review and you feel that this manuscript is now acceptable for publication, you may indicate that here to bypass the “Comments to the Author” section, enter your conflict of interest statement in the “Confidential to Editor” section, and submit your "Accept" recommendation.

Reviewer #1: (No Response)

2. Is the manuscript technically sound, and do the data support the conclusions?

Reviewer #1: (No Response)

3. Has the statistical analysis been performed appropriately and rigorously? 

Reviewer #1: (No Response)

4. Have the authors made all data underlying the findings in their manuscript fully available?

Reviewer #1: No

5. Is the manuscript presented in an intelligible fashion and written in standard English?

Reviewer #1: (No Response)

6. Review Comments to the Author

Reviewer #1: The authors have made great efforts, and the revised manuscript is ready for publication. One last comment: authors may add the justification for aiming learning style, which is a controversarial topic in education research community.

7. PLOS authors have the option to publish the peer review history of their article (what does this mean?). If published, this will include your full peer review and any attached files.

Reviewer #1: No

---

## [Editor Report · Acceptance letter]

4 May 2021

PONE-D-20-14494R3 

Differentiating the learning styles of college students in different disciplines in a college English blended learning setting 

Dear Dr. Hu:

I'm pleased to inform you that your manuscript has been deemed suitable for publication in PLOS ONE. Congratulations! Your manuscript is now with our production department. 

Kind regards, 

on behalf of

Professor Haoran Xie 

Academic Editor

PLOS ONE